# TS-ILM:Class Incremental Learning for Online Action Detection

Xiaochen Li
xiaochenli@std.uestc.edu.cn
University of Electronic
Science and Technology of
China
Chengdu, Sichuan, China

Jian Cheng*
chengjian@uestc.edu.cn
University of Electronic
Science and Technology of
China
Chengdu, Sichuan, China

Ziying Xia
zyxia@std.uestc.edu.cn
University of Electronic
Science and Technology of
China
Chengdu, Sichuan, China

Zichong Chen
chenzichonguestc@gmail.com
University of Electronic
Science and Technology of
China
Chengdu, Sichuan, China

Junhao Shi
junhaoshi@std.uestc.edu.cn
University of Electronic
Science and Technology of
China
Chengdu, Sichuan, China

Zhicheng Dong
dongzc@utibet.edu.cn
Tibet University
Lhasa, Tibet, China

Nyima Tashi
nmzx@utibet.edu.cn
Tibet University
Lhasa, Tibet, China

## Abstract

Online action detection aims to identify ongoing actions within untrimmed video streams, with extensive applications in real-life scenarios. However, in practical applications, video frames are received sequentially over time and new action categories continually emerge, giving rise to the challenge of catastrophic forgetting - a problem that remains inadequately explored. Generally, in the field of video understanding, researchers address catastrophic forgetting through class-incremental learning. Nevertheless, online action detection is based solely on historical observations, thus demanding higher temporal modeling capabilities for class-incremental learning methods. In this paper, we conceptualize this task as Class-Incremental Online Action Detection (CIOAD) and propose a novel framework, TS-ILM, to address it. Specifically, TS-ILM consists of two components: task-level temporal pattern extractor and temporal-sensitive exemplar selector. The former extracts the temporal patterns of actions in different tasks and saves them, allowing the data to be comprehensively observed on a temporal level before it is input into the backbone. The latter selects a set of frames with the highest causal relevance and minimum information redundancy for subsequent replay, enabling the model to learn the temporal information of previous tasks more effectively. We benchmark our approach against SoTA class-incremental learning methods applied in the image and video domains on THUMOS'14 and TVSeries datasets. Our method outperforms the previous approaches.

## CCS Concepts

• **Computing methodologies → Activity recognition and understanding**.

---
*Corresponding author
---

## Keywords

Online action detection, Class incremental learning

**ACM Reference Format:**
Xiaochen Li, Jian Cheng, Ziying Xia, Zichong Chen, Junhao Shi, Zhicheng Dong, and Nyima Tashi. 2024. TS-ILM:Class Incremental Learning for Online Action Detection. In *Proceedings of the 32nd ACM International Conference on Multimedia (MM '24), October 28–November 1, 2024, Melbourne, VIC, Australia.* ACM, New York, NY, USA, 10 pages. https://doi.org/10.1145/3664647.3681456

## 1 Introduction

Online action detection (OAD) aims to identify ongoing actions in video streams without foreknowledge of the future. This task holds significant sway across various real-life applications, including but not limited to autonomous driving [22, 24], video surveillance [1, 20], and anomaly detection [34, 36]. Researchers in related fields have also devised numerous efficacious solutions to tackle this challenge [8, 9, 13, 14, 28].

However, video frames are received sequentially over time in most real-world scenarios, and new action categories continually emerge. Constrained by memory and privacy considerations, the model can only access the current data, with previously observed classes being unavailable or partially accessible. Under these circumstances, a straightforward resolution is to fine-tune the mode sequentially, enabling the model to assimilate knowledge from the continuously arriving new data and categories. Nevertheless, this approach may result in the model overfitting to the current categories, leading to a marked decline in the recognition capabilities for previously learned categories (a phenomenon known as catastrophic forgetting [32]). This adverse characteristic has catalyzed extensive research within the domain of continual learning [2, 26, 44, 56]. Our focus lies on a specific variant of continual learning, Class-Incremental Learning (CIL), wherein labels across tasks are mutually exclusive, and the model lacks access to task IDs during inference. Prior works have delved into the CIL paradigm within the context of video domains [4, 37, 39, 40, 51, 52, 61], with these studies predominantly focused on scenarios where the model has the capacity to observe the entire video at any given moment. In contrast, the challenge of OAD rests on the reliance solely on historical observations without the possibility of accessing future

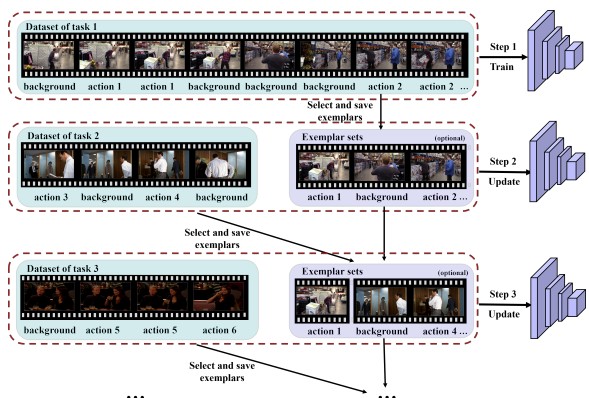

Figure 1: Illustration of CIOAD. The model assimilates knowledge from the newly acquired data at each incremental step. Concurrently, it is imperative that the model sustains its ability to recognize previously encountered data.

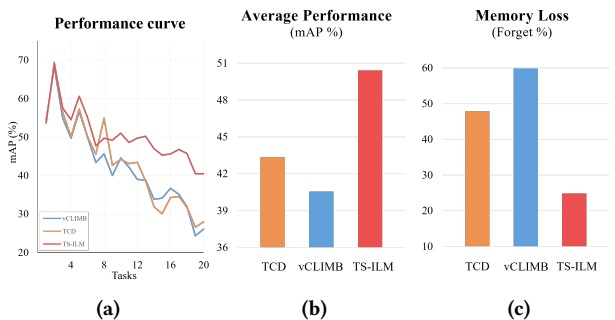

Figure 2: Comparision between TS-ILM and the SoTA CIL methods applied in video domains: (a) Comparision of the performance at each incremental step; (b) Comparision of the average performance; (c) Comparision of the memory loss on THUMOS'14 with 20 steps. The results indicate that existing methods cannot effectively solve the CIOAD task.

video frames. This restriction places a more significant demand on the model's capacity for long-duration modeling and causal reasoning and leads to an exacerbated occurrence of catastrophic forgetting. This paper first presents this problem and designates it as Class-Incremental Online Action Detection (CIOAD) (as shown in Figure 1).

Within the setup of CIL, previous methodologies have primarily tackled the issue of forgetting through two strategies [7, 46, 57]: (1) preserving old knowledge whilst acquiring new information, which typically involves limited expansion of the network architecture, and (2) employing a restrained memory budget to select and store representative samples from old classes for rehearsal. However, prior approaches tend to disrupt temporal relations to a certain extent by storing additional frames, which mitigates their effectiveness in CIOAD (as illustrated in Figure 2). To address these issues, we introduce a Time-sensitive Incremental Learning Method (TS-ILM). This method is comprised of two components: Task-Level Temporal Pattern Extractor (TPE) and Temporal-Sensitive Exemplar Selector (TES).

First, given that each action has a different temporal pattern, the backbone network might only concentrate on the temporal patterns of the action class of the current task during training, inadvertently overlooking those from past tasks. This could result in the network developing a temporal bias, leading to a decline in its performance on previous tasks. In response to this challenge, we propose the TPE, engineered to extract and preserve the temporal patterns of actions across different tasks, thereby facilitating an exhaustive temporal assessment of data prior to its entry into the backbone network.

Second, video streams are composed of a sequence of background and action frames, among which there exists a simultaneous existence of causality and information redundancy. The selection of exemplary representative samples is paramount [6, 23]. To address this, we propose TES to select and preserve a set of frames that maximize causal relationships while minimizing information redundancy on the temporal level, allowing the network to more effectively learn temporal information from previous category samples during exemplar replays.

To obtain a more compelling evaluation, we apply state-of-the-art CIL methodologies from the image and video domains to this task, establishing a baseline for the CIOAD challenge and contrasting our approach with them. We adapted the THUMOS'14 and TVSeries datasets—both frequently employed for OAD task evaluations—to align with the CIL settings. Subsequent thorough evaluations across various configurations on these adapted datasets demonstrate the effectiveness of the proposed framework.

The contributions of this paper can be summarized as follows:

- We observe that models designed for OAD experience catastrophic forgetting seriously when actually deployed. Furthermore, due to the task's need for strong temporal modeling, CIL methods applied in video domains fail to address the problem adequately. We are the first to propose this issue and name it Class-Incremental Online Action Detection.
- We propose a novel method named TS-ILM to tackle the new task. Specifically, it comprises two critical components: TPE and TES. The former extracts and preserves temporal patterns of actions corresponding to different tasks, while the latter selects a set of frames with maximal causal relevance and minimal information redundancy for subsequent replay.
- We benchmark our approach against SOTA CIL approaches applied in the image and video domains on two widely used OAD datasets that comply with the CIL settings. Our method significantly outperforms the previous approaches.

## 2 Related Work

In this section, we present methods related to CIOAD, which cover image class-incremental learning, video class-incremental learning, and online action detection.

### 2.1 Image Class-Incremental Learning

The problem of class-incremental learning has received widespread research attention in recent years and has been extensively applied within the image domain [10, 31, 35]. For image classification, current methods can be broadly divided into three categories: parameter isolation methods, regularization methods, and rehearsal methods. Parameter isolation methods alleviate catastrophic forgetting by segregating the parameter space of the neural network.

Some of these approaches dynamically change the network architecture to adapt to new information [3, 12, 45], while others implement parameter isolation by masking individual parameters or layers [29, 30]. Rehearsal methods typically involve the preservation of a representative subset of original training data [41, 44, 54, 56] or the use of generative models to emulate the training data of previous tasks [33, 47, 60]. These examples are subsequently used as supplementary data to guide the model while training new tasks, thus mitigating catastrophic forgetting to some extent. The methods based on regularization attempt to protect old knowledge from being overwritten by new knowledge by penalizing drastic changes in the weights related to previous tasks. Among these methods, some measure the importance of each parameter in the network and regularize the weights [2, 26, 27], while others focus on using regularization terms to avoid forgetting the feature representations of previous tasks [43, 49].

## 2.2 Video Class-Incremental Learning

Recent works have started to explore the problem of class-incremental learning within the video domain. [61] mitigates catastrophic forgetting by decomposing and transferring spatiotemporal knowledge. TCD [37] identifies the most contributory channel subsets within feature maps through an importance mask. vCLIMB [52] introduces a temporal consistency regularization to diminish the influence of subsampled instances on the model. FrameMaker [39] compresses videos into a single frame to conserve memory. SMILE [4] posits that in class-incremental learning tasks for action recognition, storing a single frame per video to ensure sample diversity is more efficacious than retaining the temporal information of whole videos. However, within CIOAD tasks, we are required to perform frame-by-frame recognition of actions in untrimmed videos without prior knowledge of future information, which demands greater temporal precision compared to the recognition of actions in pre-trimmed videos. TS-ILM enhances performance by extracting and saving temporal modalities of actions from various tasks, as well as by choosing and saving downsampled videos that demonstrate the strongest causal relationships and the least information redundancy on a temporal level.

## 2.3 Online Action Detection

Distinct from action recognition tasks with the luxury of observing entire videos, the objective of OAD tasks is to identify the class of the current frame without foresight. Hence, researchers concentrate on employing robust temporal modeling to address this conundrum. TRN [58] employs the latent states of LSTM [15] to model the historical and contextual relationships of videos in order to predict future actions and detect current actions via the temporal correlation between future and current actions. OadTR [55] leverages the long sequence modeling capability of Transformers [50] to capture historical data while concurrently modeling extensive global temporal information, thereby forecasting future contexts to identify the present action. LSTR [59] has instituted a mechanism for long-term and short-term memory for protracted sequence data, enabling the analysis of more historical video content. HCM [28] identifies the suitable division between action and background clips through deep metric learning [21], thus distinguishing action frames from the background ones better. Although researchers have proposed many solutions for the OAD task, how to address the catastrophic forgetting issue that arises in practical applications due to the continuous influx of data has not yet been actively explored.

## 3 Method

In this section, we first illustrate the formulation of CIOAD (Sec. 3.1), and then outline the framework of TS-ILM in Sec. 3.2. Following this, we propose the Task-Level Temporal Pattern Extractor (Sec. 3.3) to ensure a comprehensive temporal-level observation of data before it is input into the backbone network. Concurrently, we introduce the Temporal-Sensitive Exemplar Selector (Sec. 3.4) to select a set of frames that have the most significant temporal causality and minimal information redundancy for example replay. Finally, in (Sec. 3.5), we thoroughly detail the specifics of using TS-ILM for class incremental training.

## 3.1 Problem Formulation

Similar to vCLIMB benchmark [52], CIOAD requires the training of a neural network $f_\Theta : X \to Y$, which is moderated by the parameter $\Theta$. The network's objective is to discern the class $y_0$ of the current frame $x_0$ within video stream $x = [x_t]_{t=-T+1}^0$, where $\mathbf{x}_i \in X$ and $\mathbf{y}_i \in Y \ \forall i$, in the case that future frames $x_1, x_2, ...$, are not accessible. The parameter $\Theta$ of the model is learned from a sequence of m tasks, collectively referred to as $\{\mathcal{T}^1, \mathcal{T}^2, \ldots, \mathcal{T}^k, \ldots, \mathcal{T}^m\}$, where each $\mathcal{T}^k$ is associated with a unique dataset $D^k = \{V_1^k, V_2^k, \ldots, V_n^k\}$. These datasets are composed of untrimmed videos $V_i^k$, which incorporate a series of frames $\left\{\left(x_{i_1}^k, y_{i_1}^k\right), \left(x_{i_2}^k, y_{i_2}^k\right), \ldots, \left(x_{i_T}^k, y_{i_T}^k\right)\right\}$, where $y_{i_n}^k \in \{0\} \cup \left\{C_1^k, \ldots, C_{q-1}^k\right\} \subset Y$. Herein, label 0 reflects a quiescent background class devoid of action, whereas $\left\{C_1^k, \ldots, C_{q-1}^k\right\}$ corresponds to the specific action categories germane to the extant task and $q$ represents the number of categories in the task. It is worth emphasizing that in this arrangement, the action categories for each task do not overlap.

## 3.2 Overview

Figure 3 illustrates the overall framework of our method. In incremental step $k$, given an input video $V_i^k \in D^k$, following the convention [28, 55, 59], we first use the temporal feature extractor to extract the motion information of video frames, which are then concatenated with the appearance information extracted by the spatial feature extractor to form the feature sequence $F_i^k \in \mathbb{R}^{T \times D}$. Then, we input the feature sequence into Task-Level Temporal Pattern Extractor(Sec. 3.3) to ensure it is thoroughly observed on a temporal level before entering the backbone network, avoiding temporal attention bias. Afterward, we input the output $X_i^k \in \mathbb{R}^{T \times D}$ from the Task-Level Temporal Pattern Extractor into the backbone to identify the action of the current frame.

After incremental step k, given a video $V_j^k \in D^k$, we utilize the Temporal-Sensitive Exemplar Selector (Sec. 3.4) at a specific ratio to filter out a set of frames $V_j^{k''}$ to be stored in memory bank $M^k$. This set of frames maximizes causal relationships on the temporal level and minimizes information redundancy. Replaying exemplars from $M^k$ in subsequent incremental steps allows the model to better learn the temporal information of previous tasks.

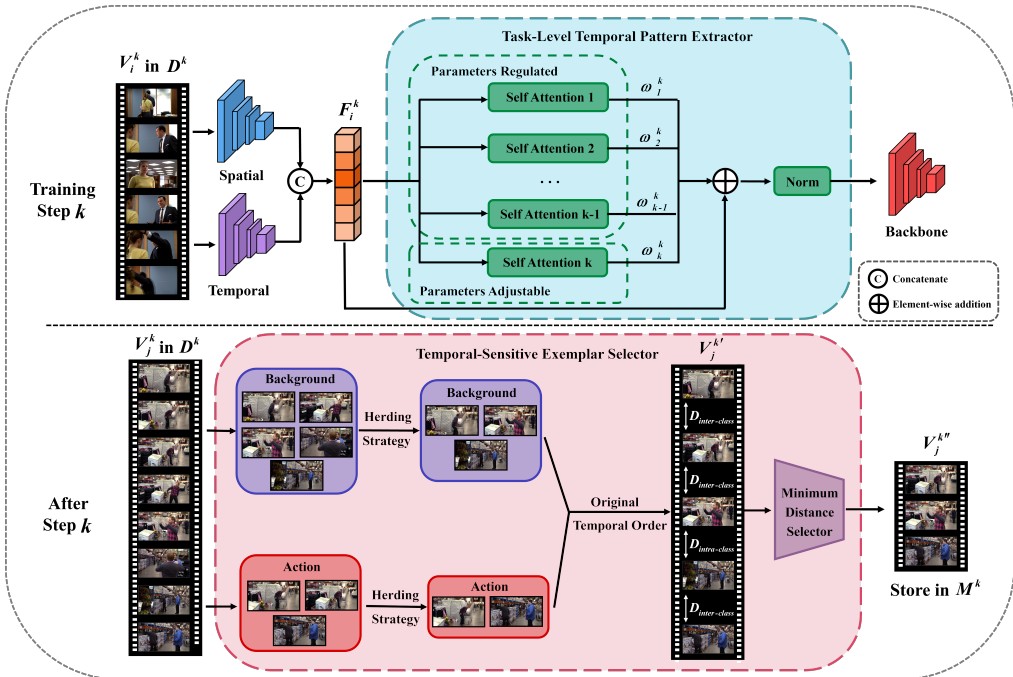

**Figure 3: An overview of the proposed TS-ILM. At each training section, we extract the temporal patterns associated with each task through the self-attention layers that belong to the respective task. The features enhanced through different patterns are then weighted and summed before being added to the original features and fed into the backbone. After training, we use the herding strategy to roughly filter out representative frames. Then, we measure the distance between frames using the inter-class distance and intra-class distance and employ a Minimum Frame Distance Selector to filter and save the set of frames with the smallest total distance for subsequent replay.**

## 3.3 Task-Level Temporal Pattern Extractor

To prevent catastrophic forgetting, we extract the unique temporal information of each task through the Task-Level Temporal Pattern Extractor (TPE) and preserve it for use during the training of subsequent tasks. This enables videos to be thoroughly observed on a temporal level before they are put into the backbone network. Formally, for each task $\mathcal{T}^k$, we design a self-attention layer $Attention^k(F_i^k; \varphi^k)$, which is a Standard Self-Attention layer[50], regulated by a weight matrix $\varphi^k$. This layer is deployed to identify which temporal segments should be paid more attention to within this task. Given the feature sequence $F_i^k$ extracted from $V_i^k \in D^k$, the output after passing through the self-attention layer associated with the task $k$ can be elucidated as follows:

$$I_k^i = Attention^k(F_i^k; \varphi^k) \in \mathbb{R}^{T \times D} \tag{1}$$

where $I_i^k$ represents the output of $k$-th self-attention layer, $F_i^k \in \mathbb{R}^{T \times D}$ denotes the input feature sequence, $T$ is the length of the feature sequence, and $D$ corresponds to the dimension of each feature. After the training of $\mathcal{T}^k$ is completed, the parameters of $Attention^k$ are frozen. Subsequently, a new self-attention layer with adjustable parameters is set to focus on the temporal information of the forthcoming task. In addition, we define a learnable weight for each self-attention layer to integrate the features output from each self-attention layer. This integration allows the fused

feature to encompass the temporal modalities of various tasks comprehensively. The sequence of weights up to $\mathcal{T}^k$ is denoted by $W = \{\omega_1, \omega_2, \ldots \omega_k\}$, and the integrated feature sequence can be calculated as follows:

$$H_i^k = \sum_{j=1}^{k} \frac{e^{\omega_j}}{\sum_{j=1}^{k} e^{\omega_j}} I_i^j \in \mathbb{R}^{T \times D} \tag{2}$$

To circumvent interference with previously preserved information by the temporal modalities of subsequent tasks during training, we introduce a regularization term designed to penalize changes in the weight parameters of self-attention layers from prior tasks:

$$\mathcal{L}_{reg}^k = \sum_{j=1}^{k-1} \left\| \omega_j^k - \omega_j^{k-1} \right\|_F^2 \tag{3}$$

where $\omega_j^k$ represents the weight parameters of the self-attention layer for task $j$ at the incremental training step $k$, and $|| \cdot ||$ denotes $\ell_2$-normalization. Following the inspiration from [16], we employ a method of residual summation to amalgamate the integrated features with the original features. This process enhances the representation capability of the features and improves training stability. Subsequently, we normalize to generate the final feature $X_i^k$, which is then input into the backbone network. Generally, this process can be formulated as follows:

$$X_i^k = \text{Norm}\left(F_i^k + H_i^k\right) \in \mathbb{R}^{T \times D} \quad (4)$$

We can learn an effective feature representation that integrates the temporal modalities of past tasks and the current task. This allows a comprehensive observation of the video before it is input into the backbone network.

## 3.4 Temporal-Sensitive Exemplar Selector

In addition, untrimmed videos contain both background frames and action frames. There might be a dense overlap of information between successive frames of the same type, which leads to a significant amount of redundant information. If we store these frames simultaneously, it cannot make effective use of the replay memory. Meanwhile, there exists a strong causal relationship in transitions between adjacent frames of different types, such as from background to action or from action to the background, which also contains a substantial amount of information. Therefore, we utilize the Temporal-Sensitive Exemplar Selector (TES) to maximize the temporal causality between frames of stored videos while simultaneously minimizing their informational redundancy. This enables the network to better learn the temporal information of past category samples through a replay mechanism. Formally, Given a video $V_j^k \in D^k \ \forall j$ after incremental step $k$, we classify each frame into two collections based on their labels: action frames set $A_j^k$ and background frames set $B_j^k$, where all frames in $A_j^k$ have non-zero labels, and all frames in $B_j^k$ have labels of zero. At a considerable ratio $\alpha$, we select respective representative subsets of frames for $A_j^k$ and $B_j^k$ using the herding strategy [44], and arrange them in their original temporal sequence $V_j^{k'} = \left\{ v_{j_1}^{k'}, v_{j_2}^{k'}, \ldots, v_{j(a+b) \times \alpha}^{k'} \right\}$ to form a temporally downsampled version of $V_j^k$, where $a$ and $b$ represent the number of frames of $A_j^k$ and $B_j^k$ respectively. Afterwards, we will further refine the selection on $V_j^{k'}$ to ensure that the ultimately selected frames have a strong temporal association. Specifically, we have defined the inter-class distance as:

$$D_{inter-class}\left(v_{j_n}^{k'}, v_{j_m}^{k'}\right) = \left\| v_{j_n}^{k'} - v_{j_m}^{k'} \right\|_F^2 \quad (5)$$

where $v_{j_n}^{k'}$ and $v_{j_m}^{k'}$ are of the same class, that is, both are either background frames or action frames, and $\|\cdot\|$ represents $\ell_2$-normalization. In addition, we define the intra-class distance as:

$$D_{intra-class}\left(v_{j_n}^{k'}, v_{j_m}^{k'}\right) = F_{\max} - \left\| v_{j_n}^{k'} - v_{j_m}^{k'} \right\|_F^2 \quad (6)$$

where $v_{j_n}^{k'}$ and $v_{j_m}^{k'}$ are of different classes, meaning one frame is a background frame and the other one is an action frame, $\|\cdot\|$ represents $\ell_2$-normalization and $F_{max}$ represents the maximum $\ell_2$-normalization between frames. Then, at a ratio $\beta$, we use the Prim algorithm [42] to select a set of frames $V_j^{k''}$ with the minimum distance from $V_j^{k'}$ for storage to replay in subsequent training. Frames of the same class in this set are spatially the farthest apart and contain the maximal amount of diverse information, thus maximizing the total information content of the frame subset. Meanwhile,

frames of different classes are spatially the closest, and the correlation between the background frame and the action frame achieves its best, thereby strengthening the causal relationship between frames.

## 3.5 Training and inference

During the training of incremental step $k$ , we use dataset $D'^k = D^k \bigcup M^{1:(k-1)}$ to update model $f_{\theta_{k-1}}$, where $D^k$ is the dataset for $\mathcal{T}^k$ , consisting of untrimmed videos that belong to $L^k$. $M^{1:(k-1)}$ is the memory bank, which is composed of the temporally downsampled videos selected by TES from $L^1$ to $L^{k-1}$, where $L^k$ represents the collection of untrimmed videos that belong to the k task, assisting the model in learning the temporal information from previous tasks better. Moreover, during the training of incremental step $k$, only the parameters of $Attention^k$ in the TPE are adjustable, while all other parameters of the self-attention layers remain frozen, with gradients set to zero. In addition, the final objective function for $\mathcal{T}^k$ is formally defined as:

$$L_{final}^k = L_{ce}^{D^k} + \gamma L_{ce}^{M^{1:k}} + \eta L_{reg}^k \quad (7)$$

where $L_{ce}^{D^k}$ and $L_{ce}^{M^{1:k}}$ respectively represent the cross-entropy loss from the training of samples for the new task data $D^k$ and the samples from the memory bank $M^{1:k}$, $L_{reg}^k$ is the regularization loss on weights in the TPE module, and $\gamma$ and $\eta$ are weights used to balance the different loss terms.

During inference, for each new video, we first use the temporal feature extractor to extract motion information from the video frames. Then, we concatenate it with the appearance information extracted by the spatial feature extractor to form a feature sequence. Subsequently, we input the feature sequence into the TPE module to ensure a thorough observation at the temporal level before entering the backbone, to avoid temporal attention bias. After that, we input the output of the TPE into the backbone to recognize the action in the current frame.

## 4 Experiments

In this section, we apply state-of-the-art CIL methods from the image and video domains to this task, constructing a baseline for the CIOAD task and comparing our method against them. We also present ablation experiments demonstrating the effectiveness of different components and analyze the results under different practical designs. Finally, we analyze the generalization ability of the model and conducted a qualitative analysis of its performance.

## 4.1 Experiment setup

**Datasets** We evaluate our model on two publicly available datasets, which are the standard OAD datasets: THUMOS'14 [18] and TVSeries [9]. The THUMOS'14 dataset consists of a training set, a test set, and a validation set, containing background frames and 20 types of action frames. We use its validation set for training and its testing set for evaluation. The TVSeries dataset is composed of 27 untrimmed long videos with actions labeled under 30 categories, with the remaining parts corresponding to the background class. Following the conventions [5, 55], we select 20 videos for training and use the remaining 7 for evaluation.

**Table 1: Comparison with the state-of-the-art approaches over online action detection class-incremental performance on THUMOS'14 and TVSeries. Our TS-ILM achieves the best performance under all experimental settings. The bold-faced numbers indicate the best performance.**

| Model | Mem. Frame Instances(G) | THUMOS'14 | | | | Mem. Video Instances(G) | TVSeries | | | |
|---|---|---|---|---|---|---|---|---|---|---|
| | | 10 Tasks | | 20 Tasks | | | 10 Tasks | | 30 Tasks | |
| | | mAP↑ | Forget↓ | mAP↑ | Forget↓ | | cAP↑ | Forget↓ | cAP↑ | Forget↓ |
| Finetuning | None | 41.27% | 68.53% | 32.76% | 74.05% | None | 71.18% | 21.02% | 63.85% | 25.73% |
| MAS | None | 30.05% | 84.37% | 34.07% | 70.20% | None | 76.59% | 14.44% | 65.92% | 31.95% |
| EWC | None | 46.95% | 55.89% | 36.14% | 64.82% | None | 75.78% | 12.75% | 65.50% | 33.09% |
| iCaRL | 0.20 | 44.01% | 65.32% | 39.97% | 58.49% | 0.17 | 75.23% | 18.13% | 65.58% | 27.02% |
| BiC | 0.20 | 46.42% | 54.74% | 42.07% | 52.92% | 0.17 | 75.51% | 16.40% | 66.32% | 24.56% |
| vCLIMB(iCaRL+TC) | 0.20 | 46.88% | 60.85% | 40.56% | 59.91% | 0.17 | 75.54% | 16.50% | 65.71% | 26.29% |
| vCLIMB(BiC+TC) | 0.20 | 48.06% | 51.75% | 42.49% | 52.16% | 0.17 | 73.88% | 17.89% | 66.17% | 23.41% |
| TCD | 0.20 | 47.83% | 55.45% | 43.34% | 47.80% | 0.17 | 72.79% | 19.24% | 66.09% | 25.48% |
| TS-ILM(ours) | 0.20 | **54.03%** | **33.13%** | **50.42%** | **24.85%** | 0.17 | **77.26%** | **13.63%** | **68.51%** | **21.46%** |
| Upper bound | 4.08 | 73.38% | - | 73.38% | - | 3.45 | 84.99% | - | 84.99% | - |

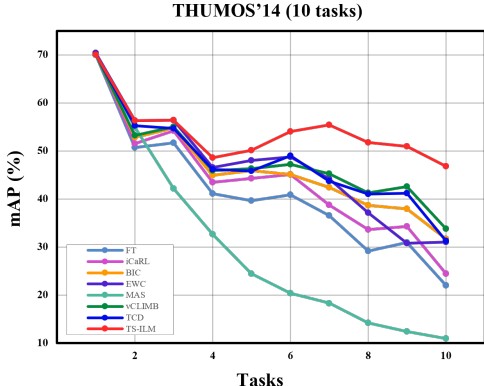

**Figure 4: The performance of various methods on THU-MOS'14 with 10 steps at each incremental step. In most incremental steps, TS-ILM achieved higher accuracy, indicating its strong capability to preserve past knowledge.**

**Benchmark** We follow the basic guidelines of CIL [11, 17], and in line with the setup for background in object detection incremental learning [48], we divide the OAD dataset into settings conforming to CIL and establish a benchmark. Specifically, for the training set, we consider action frames not belonging to the current task's categories as background frames and discard untrimmed videos that only consist of background frames, which are not used in the training of the current task. For the test set, we keep frames of all action categories that have appeared up to the current task and consider other frames as background frames. For the THUMOS'14 dataset, we split the 20 action categories into 10 and 20 tasks. For the TVSeries dataset, we split the 30 action categories into 10 and 30 tasks. Training and evaluation were conducted across all four scenarios.

**Evaluation Protocal** Following the OAD task conventions [28, 55], we assess the single incremental step of the THUMOS'14 and TVSeries datasets using per-frame mean average precision (mAP)

**Table 2: The results of using different networks as the backbone on THUMOS'14 with 10 steps.**

| Model | Backbone | mAP↑ | Forget↓ |
|---|---|---|---|
| iCaRL | LSTR | 44.01% | 65.32% |
| BiC | LSTR | 46.42% | 54.74% |
| TS-ILM(ours) | LSTR | **54.03%** | **33.13%** |
| iCaRL | TRN | 27.76% | 67.21% |
| BiC | TRN | 28.83% | 71.82% |
| TS-ILM(ours) | TRN | **31.64%** | **49.10%** |
| iCaRL | OadTR | 39.40% | 66.44% |
| BiC | OadTR | 38.91% | 60.46% |
| TS-ILM(ours) | OadTR | **45.40%** | **44.60%** |

**Table 3: Ablations for Task-Level Temporal Pattern Extractor (TPE) and Temporal-Sensitive Exemplar Selector (TES) on THUMOS'14 with 10 steps and TVSeries with 10 steps.**

| TPE | TES | THUMOS'14 | | TVSeries | |
|---|---|---|---|---|---|
| | | mAP↑ | Forget↓ | cAP↑ | Forget↓ |
| ✗ | ✗ | 41.27% | 68.53% | 71.18% | 21.02% |
| ✓ | ✗ | 43.71% | 60.33% | 72.70% | 19.20% |
| ✗ | ✓ | 52.84% | 34.73% | 76.31% | 13.25% |
| ✓ | ✓ | **54.03%** | **33.13%** | **77.26%** | **11.71%** |

and per-frame mean calibrated average precision (cAP) [9], respectively. The calibrated average precision can be formulated as:

$$cPrec = \frac{TP}{TP + \frac{FP}{w}} \tag{8}$$

$$cAP = \frac{\sum_t cPrec(t) \times I(t)}{\sum TP} \tag{9}$$

**Table 4: Ablations for memory budget on THUMOS'14 with 10 steps and TVSeries with 10 steps.**

| Model | THUMOS'14 | | | TVSeries | | |
|---|---|---|---|---|---|---|
| | Mem. Frame Instances(G) | mAP ↑ | Forget ↓ | Mem. Frame Instances(G) | cAP ↑ | Forget ↓ |
| iCaRL | 0.12 | 42.04% | 70.13% | 0.10 | 74.31% | 18.44% |
| BiC | 0.12 | 44.03% | 58.92% | 0.10 | 74.68% | 18.53% |
| TS-ILM(ours) | 0.12 | **47.74%** | **51.51%** | 0.10 | **75.76%** | **14.87%** |
| iCaRL | 0.20 | 44.01% | 65.32% | 0.17 | 75.23% | 18.13% |
| BiC | 0.20 | 46.42% | 54.74% | 0.17 | 75.51% | 16.40% |
| TS-ILM(ours) | 0.20 | **54.03%** | **33.13%** | 0.17 | **77.26%** | **11.71%** |
| iCaRL | 0.41 | 45.41% | 62.13% | 0.35 | 75.61% | 17.80% |
| BiC | 0.41 | 47.66% | 54.61% | 0.35 | 77.12% | 17.60% |
| TS-ILM(ours) | 0.41 | **56.73%** | **27.44%** | 0.35 | **79.37%** | **10.31%** |

where $I(t)$ is equal to 1 if frame $t$ is $TP$. The coefficient $w$ is the ratio between negative and positive frames. After each incremental step, we evaluate the model on all seen classes and assess the THUMOS'14 and TVSeries datasets separately through the average mAP and average cAP across all tasks (denoted by "mAP" and "cAP"). In addition, we follow [61] and report the rate of forgetting during the incremental process and the memory overhead of the saved sample frames (denoted by "Forget" and "Mem. Frame Instances(G)").

**Implementation Details** TS-ILM was implemented on PyTorch [38] using Nvidia RTX 3090 during the training and testing phases. LSTR [59] is used as our backbone, and we follow LSTR's data pre-processing process. For feature extractors, We employ a two-stream network [53] pretrained on TSN-kinetics [5], where ResNet-50 [16] and BN-Inception [19] were used for spatial and temporal subnetworks respectively. To learn the model weights, we use an Adam optimizer with a weight decay of $5 \times 10^5$ [25]. Each incremental step in the training phase lasts 25 epochs with a batch size of 16. For a fair comparison, the training settings for all methods and backbone are the same, with the loss weight $\gamma$ and $\eta$ set to 0.95 and 0.45, respectively.

## 4.2 Main Results

This section delineates the comparative assessment of our proposed TS-ILM against existing class incremental learning methods applied in the image and video domains under multiple challenging settings on two datasets. Specifically, We test the regularization methods MAS [2] and EWC [26], exemplar replay techniques iCaRL [44] and BiC [56] in the image domain, and the class-incremental learning approaches vCLIMB [52] and TCD [37] in the video domain, adapting all of these methods to the CIOAD task setting. For fair comparisons, all methods use the same feature extractor and backbone, with the exemplar replay methods using the same exemplar memory.

Table 1 reports the performance of various methods under different settings on the THUMOS'14 and TVSeries datasets, from which we can draw the following conclusions. First, TS-ILM significantly outperforms the other methods, particularly in the 20-task setting on the THUMOS'14 dataset, with the mAP increase of over SOTA by 7.08 % while reducing the forgetting rate by 22.95%, substantially mitigating the catastrophic forgetting issue in OAD tasks. Second,

**Table 5: Ablations for regularization loss on THUMOS'14 with 10 steps and TVSeries with 10 steps.**

| Regularization loss | THUMOS'14 | | TVSereis | |
|---|---|---|---|---|
| | mAP ↑ | Forget ↓ | cAP ↑ | Forget ↓ |
| ✗ | 51.78% | 35.98% | 74.20% | 19.86% |
| ✓ | **54.03%** | **33.13%** | **77.26%** | **11.71%** |

**Table 6: Ablations for each component in TES on THUMOS'14 with 10 steps and TVSeries with 10 steps.**

| Herding Stategy | Minimun Distance Selector | THUMOS'14 | | TVSeries | |
|---|---|---|---|---|---|
| | | mAP ↑ | Forget ↓ | cAP ↑ | Forget ↓ |
| ✗ | ✗ | 42.23% | 67.26% | 72.65% | 19.52% |
| ✓ | ✗ | 45.13% | 63.03% | 75.90% | 15.67% |
| ✗ | ✓ | 49.75% | 45.59% | 74.98% | 17.13% |
| ✓ | ✓ | **54.03%** | **33.13%** | **77.26%** | **11.71%** |

we find that the catastrophic forgetting on the THUMOS'14 dataset is generally more severe than on TVSeries, which we speculate is due to its more complex action patterns and greater likelihood of confusion between actions. Third, we observe that on the TVSeries dataset, storing only 5% of the data, our method reached 90.91% of the performance of joint training, i.e., the upper bound, indicating that our method utilizes replay memory effectively and significantly reduces the demand for memory capacity.

Figure 4 shows the performance of various models at each incremental step. TS-ILM achieves higher accuracy in most incremental steps, indicating its robust capability to preserve past knowledge.

## 4.3 Generalization

The proposed method specifically focuses on the issue of catastrophic forgetting that occurs during the continual learning process and is agnostic to the type of backbone, thus allowing TS-ILM to be easily integrated into any OAD method. We evaluate the generalization ability of TS-ILM using three different backbones, as shown in Table 2. The results indicate that across the three different backbones, the performance of our method consistently surpasses that of other class-incremental learning methods,demonstrating

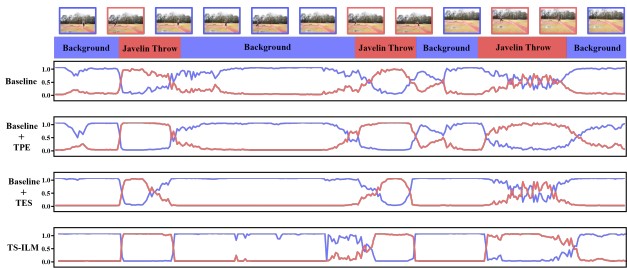

**Figure 5: Qualitative analysis of the different combinations of the two components of the proposed TS-ILM method. The bars in different colors represent the true categories, while the lines indicate the action scores of the method.**

that the superior performance of TS-ILM on CIOAD tasks is independent of the backbone category, evidencing robust generalization capabilities.

## 4.4 Ablation Study

**Effect of each component** To demonstrate the effectiveness of the Task-Level Temporal Pattern Extractor (TPE) and Temporal-Sensitive Exemplar Selector (TES), we conducted experiments to test various combinations of these two modules on the THUMOS'14 and TVSeries datasets. The experimental results for each combination are presented in Table 3, which indicates that each introduced component contributes positively to the performance, and their amalgamation yields the best performance.

**Effect of memory size** To substantiate the robustness of TS-ILM in memory consumption, we assessed the performance of various methods under different memory capacities on two datasets, with the results delineated in Table 4. It is observable that on the THUMOS'14 dataset, our TS-ILM incurs merely 0.12GB of memory expenditure, which is comparable to the performance of BiC [56] that utilizes 0.41GB, thereby economizing 70.7% of memory under equivalent performance metrics. This underlines the efficient utilization of memory by our method. Additionally, the table reveals that regardless of the memory budget, TS-ILM consistently surpasses other methodologies, indicating its outstanding capability to counteract catastrophic forgetting.

**Effect of regularization loss** As discussed in Section 3.3, within the TPE module, we define a learnable weight for the self-attention layer associated with each task, denoted by $\omega$, and introduce a regularization loss $L_{reg}$ to penalize variations in the weight parameters of the self-attention layers that pertain to prior tasks. To evaluate the efficacy of this loss, we assessed performance with and without the loss on two datasets, as shown in Table 5. The results indicate that this additional loss has led to performance gains of 2.25% on THUMOS'14 and 3.06% on TVSeries, while the forgetting rate decreased by 2.85% and 8.15% respectively. These findings suggest that this loss function can effectively preserve information from past actions and mitigate catastrophic forgetting.

**Effect of each component in TES** As described in Section. 3.4, our TES module can be roughly decomposed into two components: initially, frames are coarsely filtered using the herding strategy [44], and then further refined and stored through Minimum Distance Selector. Table 6 enumerates the results of testing different combinations of the two parts of TES on two datasets. It is noteworthy

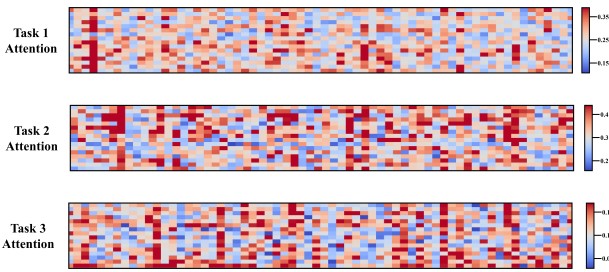

**Figure 6: Attention visualization maps. It demonstrates how the self-attention heads, belonging to different tasks, focus differently on the temporal dimension. The colorbar indicates the degree of attention to time.**

that the tests were conducted with the same memory budget and that a random sampling strategy was employed to select the frames to be saved when both components were disabled. The findings corroborate that each component individually contributes to performance improvement, and their conjunctive application leads to superior performance outcomes.

## 4.5 Qualitative Analysis

**Visualization of action scores** Figure 5 visualizes video clips and their corresponding action scores. These action scores were inferred from videos in the dataset of the 10-th task after training on 10 tasks on THUMOS'14. The category "Javelin Throw" displayed in the figure is part of the training set of the 7-th task and did not appear in subsequent training. The results indicate that both components of our proposed TS-ILM, as well as their combination, significantly enhance the model's ability to remember previous action categories, effectively overcoming the issue of catastrophic forgetting.

**Visualization of self-attention layers in TPE** As detailed in Section 3.3, within the TPE module, we integrate a self-attention layer for each task to memorize its temporal action patterns. Figure 6 displays the visualized results of the self-attention layers for different tasks, where higher values indicate greater attention by the self-attention layer at that specific time. The results suggest that the focus of self-attention layers varies across different tasks. Integrating these observations can effectively prevent biases in the backbone of the temporal dimension caused by discrepancies in the categories of actions in different tasks.

## 5 Conclusion

In this paper, we introduce the novel task of Class-Incremental Online Action Detection and propose an innovative framework to address this challenge. Specifically, our framework comprises two key components. The first one extracts and saves the temporal patterns of divergent actions in different tasks, allowing for a comprehensive temporal analysis before the data enters the backbone network. The second one selects a set of frames that maximize temporal causality and minimize information redundancy for subsequent replay, enabling the model to learn the temporal information of previous tasks better. Finally, we establish a benchmark for this task and conduct a thorough evaluation of our approach in comparison to state-of-the-art class-incremental learning methods previously applied in the image and video domains, demonstrating the effectiveness of our method.

## 6 Acknowledgments

This work was supported in part by the NNSFC&CAAC under Grants U2233209 and U2133211, in part by the Natural Science Foundation of Sichuan, China under Grant 2023NSFSC0484, and in part by the National Natural Science Foundation of China under Grant 62071104.

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
