# OpenReview forum: "TS-ILM:Class Incremental Learning for Online Action Detection"
_acmmm.org/ACMMM/2024/Conference — MM2024 Poster_

### Official Review · Reviewer_jYDj · 2024-05-13

**Rating:** 4
**Confidence:** 3

**Summary:**

In this paper, the authors propose ClassIncremental Online Action Detection (CIOAD), which is class-incremental learning in online action detection. Due to the need for more precise temporal modeling in online action detection, the authors found that applying existing class-incremental learning algorithms to CIOAD does not yield good performance. They have introduced a framework more suitable for CIOAD, called TS-ILM. In this framework, the task-level temporal pattern extractor extracts the temporal patterns from different tasks before the data is input into the backbone, allowing for a more comprehensive observation of the data on a temporal level. The temporal-sensitive exemplar selector selects a subset of frames from the video for subsequent replay. TS-ILM has demonstrated excellent performance on several challenging benchmarks.

**Strengths:**

1. This paper introduces the task of ClassIncremental Online Action Detection (CIOAD) for the first time, which focuses on class-incremental learning within Online Action Detection (OAD) and has practical value for the application of OAD
2. The proposed TS-ILM is easy to follow and has demonstrated excellent performance on the task of ClassIncremental Online Action Detection (CIOAD)
3. Numerous ablation studies and qualitative comparisons help in understanding the setting
4. A large number of existing methods have been replicated on the task of CIOAD, and the code is easy to read

**Limitations:**

1. In Equation 3, the authors propose a regularization term. However, as shown in Figure 3, the self-attention parameters from 1 to k-1 are frozen during task k. Does this imply that the regularization term will only optimize the self-attention for k? If so, as shown in Equation 3, it calculates the regularization for 1 to k-1, but would it not be effective for optimization? Despite the results in Table 5 showing a clear effect of the regularization loss, I remain confused about this part.
2. The symbols and definitions in some parts of the article are difficult to understand. For example, what does q represent in \( C^k_{q-1} \) at line 303? And what is \( L^k \) at line 521?
3. In Online Action Detection (OAD), data is generally fed into the network frame by frame. However, as the authors have described in the text, the TS-ILM uses a two-stream network to extract features, and it is usually necessary to extract optical flow in the Temporal branch. Would this affect the response speed of OAD in practical applications?
4. The paper lacks an inference step.

**Suitability:**

2

---

### Official Review · Reviewer_CJkG · 2024-05-24

**Rating:** 4
**Confidence:** 2

**Summary:**

This paper proposes an approach to solve the task of Online Action Detection, specifically the newly introduced task of Class Incremental Online Action Detection (CIOAD). To solve this task, a new framework is proposed, namely TS-ILM. TS-ILM contains two parts: a task-level temporal pattern extractor and a temporal-sensitive exemplar selector. The temporal pattern extractor extracts the temporal
patterns of actions in different tasks and saves them. The temporal-sensitive exemplar selects frames with high relevance to learn temporal
information of previous tasks more effectively. The proposed model is evaluated on THUMOS'14 and TVSeries dataset and report improvements over the current SOTA.

**Strengths:**

--> The introduction of class incremental learning to the problem of online action detection is new.
--> The components of the proposed architecture, the task-level temporal pattern extractor, and the temporal-sensitive exemplar selector are explained well in Fig 3.
--> Ablation experiments are thorough, covering all the component and the hyper-paraments in the method.
--> The paper is well-organized and well-written. Overall it is not hard to read and understand.
--> Results on both datasets beat the baselines across different settings.

**Limitations:**

--> The motivation to combine Class Incremental Learning with Online Action Detection is not clear. In Online Action Detection, during inference, the constraint is to detect an ongoing action without accessing the future frames. I do not understand how catastrophic forgetting is relevant here.
--> Addition symbol in Fig 3, should be element-wise addition and pixel-wise addition.
--> What are the x and y-axis in the attention maps of Fig. 6?
--> It is mentioned in lines 571-574, that during evaluation, the number of tasks is set to 20 and 30 for THUMOS' and TVSeries datasets respectively. Why are these different from the number of tasks during training?
--> The THUMOS'14 and TVSeries are the only two datasets used for evaluation. Any reason for not evaluating on HDD dataset or EpicKitchen like the recent works on OAD?

**Suitability:**

1

---

### Official Review · Reviewer_2qgQ · 2024-05-28

**Rating:** 4
**Confidence:** 4

**Summary:**

This paper proposes the task of Class-Incremental Online Action Detection and introduces the TS-ILM framework to deal with this task.

**Strengths:**

(1) The analysis of Class Incremental Learning for Online Action Detection is interesting and reasonable.
(2) Empirical results are extensive and demonstrate the effectiveness of the method.

**Limitations:**

(1) For the task-level temporal pattern extractor, the design of the parameter adjustable module is not clearly illustrated. What is the motivation to add another self-attention module on top of the previous self-attention modules?

(2) For the temporal-sensitive exemplar extractor, how do the background and action be distinguished?

**Suitability:**

3

---

### Meta-Review · Area_Chair_3X4S · 2024-06-30

**Recommendation:** Accept (Poster)
**Confidence:** 4

**Metareview:**

In this work the authors focus on class-incremental learning for online action detection. It is a novel task and the authors propose a new method to solve this. This work received 3x borderline accept ratings before the rebuttal and the authors have provided a response to reviewers comments. The main concerns were lack of clarity/motivation on proposed components, motivation behind this new task, and limited datasets for evaluation. The rebuttal addressed most of the concerns and the reviewers kept their original borderline scores. The motivation behind why class-incremental learning is relevant for online action detection is still not well explained and non-multimodal aspect of this work weakens its relevance to multimedia community. Despite these limitation, introduction of this new task and extensive evaluation is a strong point of this work. Considering these points, the AC recommends acceptance of this work.